# Validation of an Empirical Subwaveform Retracking Strategy for SAR Altimetry

**Marcello Passaro** [1,*] ⓘ**, Laura Rautiainen** [2]**, Denise Dettmering** [1] ⓘ**, Marco Restano** [3]**, Michael G. Hart-Davis** [1] ⓘ**, Florian Schlembach** [1]**, Jani Särkkä** [2] ⓘ**, Felix L. Müller** [1] ⓘ**, Christian Schwatke** [1] ⓘ **and Jérôme Benveniste** [4]

1   Deutsches Geodätisches Forschungsinstitut, Technische Universität München (DGFI-TUM), 80333 München, Germany
2   Finnish Meteorological Institute, FI-00101 Helsinki, Finland
3   SERCO/ESRIN, Largo Galileo Galilei, I-00044 Frascati, Italy
4   European Space Agency-ESRIN, Largo Galileo Galilei, I-00044 Frascati, Italy
*   Correspondence: marcello.passaro@tum.de

**Abstract:** The sea level retrievals from the latest generation of radar altimeters (the SAR altimeters) are still challenging in the coastal zone and areas covered by sea ice and require a dedicated fitting (retracking) strategy for the waveforms. In the framework of the European Space Agency's Baltic + Sea Level (ESA Baltic SEAL) project, an empirical retracking strategy (ALES + SAR), including a dedicated sea state bias correction, has been designed to improve the sea level observations in the Baltic Sea, characterised by a jagged coastline and seasonal sea ice coverage, without compromising the quality of open ocean data. In this work, the performances of ALES + SAR are validated against in-situ data in the Baltic Sea. Moreover, variance, crossover differences and power spectral density of the open ocean data are evaluated on a global scale. The results show that ALES + SAR performances are of comparable quality to the ones obtained using physical-based retrackers, with relevant advantages in coastal and sea ice areas in terms of quality and quantity of the sea level data.

**Keywords:** sea level; retracking; satellite altimetry; tide gauge; Baltic Sea

## 1. Introduction

Since the early 90 s, global sea level data are routinely collected by radars onboard satellites. The variable to be estimated is the range, i.e., the distance between the satellite centre of mass and the sea surface. The precise estimation of range relies on the fitting of the signal registered in the form of time series called waveforms [1]. These are typically provided at a rate of 20 Hz (referred to as roughly 300 m along the satellite track) and the fitting of these signals is called retracking. Retracking algorithms are subdivided into functional forms derived from models that simulate the physics of the interaction between the radar signal and the illuminated surfaces (i.e., physical retrackers) and empirical algorithms that estimate parameters based on purely geometrical considerations on the retrieved signal [2].

Since Cryosat-2, most of the altimeter missions currently flying adopt the Synthetic Aperture Radar (SAR) processing, which generates multi-looked waveforms characterised by a much smaller along-track footprint [3]. While this improves the signal-to-noise ratio and the performances in the coastal zone, several studies have shown that retracking is still necessary to enhance the quality and quantity of altimetry retrievals in these challenging areas (e.g., [4]). To tackle this, in particular, the SAMOSA+ retracker was proposed as an enhancement to the physical SAMOSA2 model, which is currently the adopted baseline for SAR altimeters [5].

Empirical algorithms have been adopted to retrieve water levels from signals in which sea ice or land intrudes the satellite's footprint. Examples of their application can be found mostly for inland waters [6] and polar oceans [7]. To our knowledge, no empirical

retracking algorithm has been shown to work as well as the standard physical retrackers both in the open ocean as well as in the coastal zone.

This study presents an extensive validation of ALES + SAR, an empirical retracker that uses a simplified version of a functional form inherited by physical retrackers of previous altimeters (nowadays known as Low-Resolution Mode, LRM) to estimate sea level in the open ocean, coastal zone and sea-ice-affected areas. The retracker was developed in the framework of the ESA Baltic SEAL project to improve the availability of Sentinel-3A sea level data in SAR mode. It was used together with a multi-mission coastal altimetry dataset to revisit the regional sea level trend and variability in the Baltic Sea during the altimetry era [8]. The objective of this paper is to evaluate the ALES + SAR performances. This is done regionally in the Baltic Sea using in-situ data and a physical coastal retracker as a comparison (SAMOSA+), and globally using crossover and gridded sea level variance analysis as well as the power spectral density.

The paper is structured as follows: Section 2.1 describes the data types and sources; Sections 2.2 and 2.3 recall the main characteristics of the retracker and its sea state bias correction; Sections 2.4–2.7 describe the methodology used for validation; Section 3 presents the main findings and interprets the results.

## 2. Data and Methods

We briefly summarise the content of this section. In Section 2.1 we list the data sources associated with the different altimetry processing and with the external validation dataset. Subsequently, we describe the methodology behind the ALES + SAR data: firstly the steps that lead to the fitting of the altimetric waveform (Section 2.2) and secondly the relation adopted to correct the range estimations for the influence of the sea state, i.e., the sea state bias (SSB, Section 2.3). From Sections 2.4–2.7 we explain all the metrics that are used to analyse the performances of ALES + SAR with respect to the other datasets. Such metrics are the following: an external regional validation against in-situ data located in the Baltic Sea Section 2.4; an internal global validation based on sea level differences at crossover and variability of sea level (Sections 2.5 and 2.6); a global validation in the wavenumber domain to assess the energy content of the sea level signal sampled along the altimetry tracks (Section 2.7).

### 2.1. Data

We use Level-1B data of the Sentinel-3A mission acquired from the Copernicus Online Data Access (CODA) catalogue of EUMETSAT (https://coda.eumetsat.int/). The default retracking algorithms is an implementation of the physical, fully analytical, open ocean SAMOSA2 waveform model [5,9]. The product also contains the multi-looked SAR altimetric waveforms, which are retracked using the ALES + SAR algorithm as described in this work.

The SAMOSA2 retracker has been adapted to improve its performance in the case of multipeak and peaky waveforms, typical of sea ice and coastal zone. This improved retracker is called SAMOSA+ and is described in [5]. In this study, SAMOSA+ ranges and SSB corrections are obtained using the SARvatore for Sentinel-3 service of the European Space Agency's Altimetry Virtual Lab (AVL) platform hosted on EarthConsole (https://earthconsole.eu), using the "Sentinel-3" profile preset. It has to be noted that SAMOSA+ data were specifically processed from the ESA Altimetry Virtual Lab platform for the ESA Baltic SEAL project and therefore their availability in this study is limited to the regional analysis in the Baltic Sea.

Once the altimetric ranges (distance from the satellite to the sea level) have been obtained by retracking the waveforms, the sea surface height (called also sea level in this article) is obtained by subtracting the range and the sum of instrumental and geophysical correction from the orbital altitude. The set of geophysical corrections used in this study for all the altimetry data, as well as the waveform classification applied to distinguish returns



from leads among sea ice, is inherited from the ESA Baltic SEAL project (see [10], Table 2), whose along-track altimetry products are freely distributed at http://balticseal.eu/.

To validate the sea level data from satellite altimetry against in-situ retrievals, we use the tide gauge network in the Baltic Sea, assembled in the ESA Baltic SEAL project and described in [10]. It was assembled using data from the Copernicus Marine Environment Monitoring Service (CMEMS) database, the Danish Meteorological Institute (DMI), the Finnish Meteorological Institute (FMI) and the Swedish Meteorological and Hydrological Institute (SMHI). The dataset is referenced to a unified height system (EVRF2007) and the accuracy of a single sea-level measurement from tide gauges after quality control is within 1 cm [11].

### 2.2. Methods: Retracking

ALES + SAR is an empirical retracker. It uses a mathematical formulation to describe the typical shape of the altimetric waveform. The mathematical formulation and the methodology of fitting a subwaveform of the signal is adapted from its LRM counterparts ALES [12] and ALES+ [13], which avoid the contribution of the spurious reflections within the footprint.

In particular, ALES + SAR adopts a simplified version of the Brown-Hayne functional form to track the leading edge of the waveform. The Brown-Hayne functional form was developed to analytically resolve the convolution that describes the mean backscattered return power from a rough ocean [14,15]. This analytical form corresponds to an error function multiplied by a decaying exponential, a description that still allows to correctly derive the epoch, i.e., a given position along the leading edge with respect to the fixed nominal tracking point determined by the onboard tracker.

In the next paragraphs, the procedure followed by the ALES + SAR retracker is summarised, while Figure 1 provides three examples of fitting waveforms characterised by a response typical of the following conditions: open ocean (leading edge and decaying trailing edge), coastal ocean (trailing edge with excess power, i.e., multi-peak waveform) and leads among sea ice (peaky waveform).

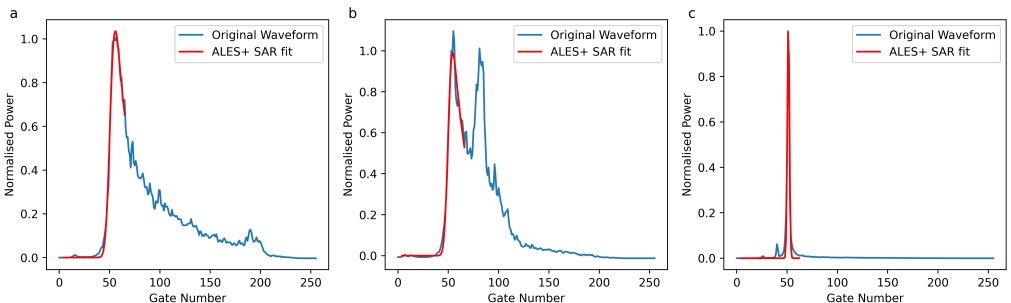

**Figure 1.** Three examples of ALES + SAR fitting applied to a SAR altimetry waveform from (**a**) typical open ocean conditions, (**b**) coastal-like interference along the trailing edge and (**c**) lead-like peaky leading edge.

### 2.2.1. Functional Form

The simplified version of the Brown-Hayne functional form used to retrack SAR waveforms is:

$$V_m(t) = P_u \frac{[1 + \mathrm{erf}(u)]}{2} \exp(-v) + T_n \tag{1}$$

where

$$u = \frac{t - \tau - c_\xi \sigma_c^2}{\sqrt{2}\sigma_c} \qquad v = c_\xi \left( t - \tau - \frac{1}{2} c_\xi \sigma_c^2 \right) \tag{2}$$

where $V_m$ is the return power, $\tau$ the epoch with respect to the nominal tracking reference point (linked to the range), $P_u$ the amplitude of the signal, $T_n$ the thermal noise level. $\sigma_c$

is the parameter affecting the rising time of the leading edge (i.e., the number of gates between its start and its end) and $c_\xi$ is the parameter affecting the exponential decay of the trailing edge.

ALES + SAR tracks the mid-point of the leading edge as the reference point to compute the range, while the physical retracker in use for SAR altimetry establishes that the retracking point is located at 84.22% of the maximum for each Doppler beam [16]. As the validation in this article will demonstrate, this does translate into a notable decrease in the performances of ALES + SAR. This is likely due to the fact that such a difference in retracking point creates a difference in sea level estimation that is dependent on sea state. However, such sea-state-related effects are captured by the empirical SSB model described in Section 2.3.

### 2.2.2. Leading Edge Detection

The procedure to automatically detect the position of the leading edge along the altimetric waveform is purely based on geometrical considerations. This follows the same methodology as the ALES+ retracker, which is designed for Low-Resolution Mode altimetry [13]. In this case, the only difference between ALES+ and ALES + SAR are the values of the thresholds used to search for the leading edge, due to the different signal-to-noise ratios. For the sake of the reproducibility of the algorithm, the interested reader can find the full procedure described in [17].

### 2.2.3. Choice of Trailing Edge Slope

As an empirical retracker, ALES + SAR is not able to physically define the slope of the trailing edge of a multi-looked waveform. Nevertheless, the trailing edge decay does not influence the fit of the leading edge for a subwaveform retracker. For ocean waveforms, we therefore use a predefined value to tune the factor driving the decay of the exponential in Equation (2), i.e., $c_\xi = 0.04$. In cases of peaky waveforms identified by the leading edge detection, $c_\xi$ is left as an unknown to be estimated by the retracker in a preliminary step before the final estimation of the range. As in the case of the leading edge detection, there is no difference compared to the ALES+ procedure described in [13].

### 2.2.4. Subwaveform Retracking

The concept of subwaveform retracking aims at fitting waveforms whose trailing edge is perturbed by areas of the footprint with different backscatter conditions, such as patches of calm waters, land, or ice, while guaranteeing a comparable accuracy as for typical open ocean conditions.

In ALES + SAR, the retracking step consists of a single pass on a subwaveform defined empirically as:

$$\text{Stopgate} = \text{StopgateLE} + 20 \tag{3}$$

where StopgateLE is the last gate of the leading edge, given an original waveform defined over 128 gates. The impact of changing the Stopgate is not investigated in this study, whose aim is to assess the performances of ALES + SAR in its original configuration.

Before proceeding with the next section describing the SSB correction, we repeat for clarity the main differences between ALES+ and ALES + SAR. ALES+ is used for LRM waveforms, therefore the Brown-Hayne functional form physically describes the interaction between the radar pulse and the ocean surface. ALES + SAR is instead an empirical retracker for SAR altimetry, because the same functional form (simplified as described above) is used to geometrically describe the waveform shape, but its parameters cannot be directly computed based on the physics of the measurement. For example, the slope of the trailing edge in the open ocean in ALES + SAR is not computed as a function of the beam width and the mispointing angle, and the significant wave height is not derived as a function of the rising time of the leading edge. Moreover, while in ALES+ the subwaveform is chosen based on a preliminary estimation of the sea state, ALES + SAR adopts a fixed subwaveform width.

### 2.3. Methods: Sea State Bias

An *SSB* model is computed parametrically for ALES + SAR using sea level residuals (with no *SSB* correction applied) at the crossover points. We use a wider region covering the North Sea and the Mediterranean Sea to have more open ocean crossover points, which are scarce in the Baltic Sea. The proportionality between the rising time of the leading edge and the significant wave height has already been shown in SAR altimetry [18]. In this case, we suppose that the *SSB* correction is directly proportional to $\sigma_{c_t}$, which is the rising time of the leading edge ($\sigma_c$) converted in metres using the equation of the two-way travel time of a radar pulse. The coefficient of proportionality, which is the unknown, is defined as $\hat{\alpha}$:

$$SSB = \hat{\alpha}\sigma_{c_t} \qquad (4)$$

where:

$$\sigma_{c_t} = 2c\sigma_c \qquad (5)$$

and *c* is the speed of light.

Considering the sea level differences ($\Delta SLA$) at the crossover locations, a set of linear equations is obtained:

$$\Delta SLA = \hat{\alpha}\Delta\sigma_{c_t} + \epsilon \qquad (6)$$

where $\epsilon$ is the residual sea level difference that is not proportional to $\sigma_{c_t}$. Equation (6) is solved for $\hat{\alpha}$ in a linear least-square sense. The chosen $\hat{\alpha}$ is the one that maximises the variance explained at the crossovers, i.e., the difference between the variance of the crossover difference before and after correcting the sea level estimations for the SSB. We found an optimal value for $\hat{\alpha} = 0.03$.

### 2.4. Methods: Comparison with Tide Gauges

The sea level of ALES + SAR was compared to the Baltic Sea tide gauge (TG) data. The time series for the comparison were formed by identifying the nearest ALES + SAR along-track points to each TG within 30 km (from the TG) and both 0–3 km and 3–10 km away from the coast. The temporal resolution is high for the majority of the Baltic Sea TGs (1 h) which allows for the data to be interpolated to the time of the altimeter overpass with one-minute accuracy.

The fitting error from ALES + SAR on the leading edge of the normalised waveform and the Sea Ice Index obtained by unsupervised open water detection was used as a quality flag for the altimeter data in ESA Baltic SEAL. For the validation, we did not correct the sea level data for the atmospheric component (the Dynamic Atmospheric Correction DAC) to guarantee the comparability with the tide gauges. Tides were not considered as the tidal variability is negligible in this region [19].

The estimations of SAMOSA2 and SAMOSA+ were also compared to the TGs. For SAMOSA2, which is not a coastal retracker, no specific quality flag is applied, and the selected points are based on the ALES + SAR quality flag previously described. For SAMOSA+, which is designed to improve coastal performances, the SAMOSA+ quality flag was used. This was defined in [5]: range retrievals are considered valid if the misfit between the power waveform model and power waveform data at 20 Hz is equal to or less than 4. Note that of course the same sea ice detection is applied to all datasets.

The statistics for the comparison are the Pearson correlation (*r*) on a confidence interval of 95%, the Root Mean Squared Error (RMSE) and the number of valid points. The confidence interval corresponds to a *p*-value of 0.05 set to exclude the null hypothesis, which in this case is that the slope of the regression between the two distributions (altimetry and TG data) is zero, using the Wald Test with t-distribution of the test statistic. If the *p*-value is greater than 0.05, the correlation is not displayed as it is rejected as statistically insignificant. The altimeter sea level anomalies that were greater than double the standard deviation were excluded from the r and RMSE analysis (i.e., considered invalid). The tide gauge sea level was detrended to remove the effect of land uplift, which is prominent, especially in

the northern Baltic Sea but missing from the altimeter sourced absolute sea level. Further quality control was performed by using TG minimum and maximum values to filter peaks in the altimeter time series. In order to compare the time series, differences in the reference frames were accounted for. Accordingly, the national height system referenced TG data was unified with the EVRF2007 reference frame. The EVRF2007 reference frame is tied to the Normaal Amsterdams Peil (NAP). As the altimeter-sourced sea level is tied to the TOPEX reference ellipsoid, the mean of the sea level was removed from the altimeter time series and set equal to the mean of the corresponding TG time series for each altimeter-TG pair.

### 2.5. Methods: Crossover Analysis

Sea surface height crossover differences are analysed globally based on the two re-tracker solutions ALES + SAR and SAMOSA2 using identical geophysical corrections, except SSB which is tuned to the respective retracker (see Section 2.3). Single-satellite, as well as dual-satellite crossovers, are analysed. Crossover differences are computed based on 1-Hz data, which are derived by building the median of 20 Hz orbit height minus range minus mean sea surface data and adding back 1-Hz orbit height and sea surface heights. The connection between the 1-Hz data and the high-frequent data is done based on the 1-Hz-flag available in the SGDR data. Since this is not available before cycle 25, the crossover analysis only starts in November 2017.

Within the single-satellite crossover (SXO) analysis, all possible sea level differences between ascending and descending passes of Sentinel-3A are built within one cycle, i.e., with a maximum time difference of 27 days. All values with differences larger than 1 m or with a standard deviation of higher than 0.1 m are considered outliers and removed from the statistics. For each Sentinel-3A cycle, the mean and standard deviations of the crossover differences are computed, as well as the mean of the standard deviations of the crossover differences. Regions north and south of 60-degree latitude have been neglected from the comparison to minimise influences from sea ice coverage.

In addition, a multi-mission crossover analysis (MMXO) is performed. Here, in addition to the Sentinel-3A SXO, sea level crossover differences with Jason-3 were computed. For this investigation, the maximum time difference between the crossing tracks is set to two days. All crossover differences are minimized in a least-squares approach to estimate radial errors for both missions (Sentinel-3A and Jason-3) following the approach published by Bosch et al. [20]. This method reveals possible inconsistencies between the missions and is particularly suitable for identifying long-term drifts in instruments or model corrections [21].

### 2.6. Methods: Gridded Sea-Level Variance Analysis

Using the same data as used in the crossover analysis, gridded sea level anomaly variance analysis (SLAVA) is conducted for ALES + SAR and SAMOSA2 retracker solutions. The SLAVA is a valuable metric that has been used to evaluate the impact that different geophysical corrections have on the estimations of sea level from satellite altimetry (see examples in [22,23]). Within this study, two variations of the SLAVA are presented to assess how the retrackers perform both globally and in the coastal regions. For both approaches, all absolute values of sea level anomaly that exceed 2.5 m are considered outliers and are removed from the analysis. Furthermore, in order to neglect errors caused by sea ice coverage in the higher latitudes, the analysis is restricted to a range between 60 degrees north and south.

The first approach follows a similar approach to Hart-Davis et al. [22], where the sea level data for each pass and cycle of Sentinel-3A is gridded onto a two-degree grid for both retracker solutions. This grid size is chosen to account for the maximum inter-track distance of Sentinel-3A, which is approximately 104 km at the equator. Using the range estimations from the individual retrackers, the variance of each grid point is calculated globally. Contrasting the resultant two global SLAVAs allows for the identification of regional differences between the retrackers as well as an assessment of the overall differences.

Instead of gridding the data into longitude/latitude points, the second approach involves mapping the sea level anomalies as a function of distance to the coast. With the focus on the performances of ALES + SAR in the coastal region, each sea level observation is then gridded into a 3-km distance to the coast grid before the standard deviation and variance of each grid point are calculated.

### 2.7. Methods: Spectral Analysis

We compare the sea level anomalies generated using ALES + SAR with the official data from Sentinel-3A, generated using SAMOSA2, also in terms of spectral content. In order to focus the evaluation on the range retrieval and SSB correction, the same exact geophysical corrections are applied to both datasets.

To have enough points, we evaluate 8 full cycles (cycles 30 to 37) over the global ocean. The points located closer than 20 km to the coast and sea level anomalies exceeding 2 m in absolute value are excluded. While these thresholds are purely empirical and aimed at excluding outliers, we note that our choice is less conservative than what has been applied in previous literature (for example [24] excluded points closer than 50 km to the coast, while [25] used 1.5 m in absolute value as threshold). The energy content at different length scales is evaluated by computing the Fourier components using the Welch periodogram method [26]: a spectrum is generated with a fast Fourier transform for every successive 1024-point segments of sea level data. The spectra of the different segments are then averaged together.

Although this analysis has been often erroneously overlooked in the previous literature concerning coastal retracking, the procedure is well-known and routinely used for quality assessment of sea level data in the open ocean (e.g., [24]). Full details of the procedure are found in [4].

## 3. Results

### 3.1. Sea State Bias

In order to understand the validity of the SSB correction derived for ALES + SAR, we analyse the amount of variance explained by the application of this correction considering the sea level differences at the crossover points. For comparison, we use the same statistics (Table 1) reported by [27], which computed a global parametric SSB correction applied on 1-Hz data from the TOPEX/POSEIDON mission. In addition, we compared them to the values obtained by [28], which computed the SSB regionally and applied it to 20-Hz data retracked with ALES applied to Jason-1 (i.e., a physical, subwaveform retracker).

Firstly, it is observed that the variances before and after the application of the SSB are smallest for ALES + SAR when compared to older studies. This confirms that the precision that can be reached with SAR altimetry is higher than in the previous missions.

Secondly, the variance explained by the SSB correction computed for ALES + SAR is exactly the same as the one explained by the correction applied to the physical ALES retracker on Jason-1 (20%). The fact that both of these explained variances are higher than in [27] is due to the well-known issue of intra-1 Hz correlation of the errors in the estimation of sea state and range [29], which is mitigated by the application of an empirically-derived SSB at 20 Hz instead of at 1 Hz [28,30].

We conclude that a simple parametric estimation of SSB based on the rising time of the leading edge with ALES + SAR is as effective as the correction based on sea state and wind estimates, which is applied to the physical retrackers of standard altimetry missions.

**Table 1.** Variance at crossover locations (XO var) before and after the application of the SSB. The first row provides the corresponding numbers reported in [27] for a global solution using 1 Hz data.

| Dataset | XO var before SSB [cm$^2$] | XO var after SSB [cm$^2$] | Variance Explained |
|---|---|---|---|
| Gaspar et al. (1994) [27] | 127.7 | 120.4 | 6% |
| SGDR Jason-1 Mediterranean Sea | 135.6 | 108.4 | 20% |
| ALES + SAR Sentinel-3A | 106.0 | 84.9 | 20% |

### 3.2. Comparison with Tide Gauges

The close proximity to the coast causes challenges for sea level estimations, which is also reflected in the results. ALES + SAR, SAMOSA2 and SAMOSA+ all performed significantly better in the 3–10 km range than within the closest 0–3 km. However, within this close proximity to the coast, ALES + SAR shows potential with *r* of 0.5 and RMSE of 35 cm and (Table 2). Although SAMOSA+ is able to produce the best correlation and RMSE in the 0–3 km range, the quality flag reduces the number of points down to half when compared to the number of points achieved using the ALES + SAR quality flag. In the 3–10 km range, ALES + SAR outperforms SAMOSA+ and SAMOSA2 within the 3–10 km range with an overall *r* of 0.7 and RMSE of 22 cm (Table 2).

We recall that, while a specific quality flag on the SAMOSA2 ranges was not applied, the comparison using SAMOSA+ uses the misfit as suggested in the previous literature. This brings a higher correlation than ALES + SAR in the 0–3 km band but comes at a significant cost in terms of the number of points. Moreover, this benefit concerns only the first 3 km, since in the band 3–10 km from the coast ALES + SAR has better statistics than SAMOSA+ despite the higher number of points considered. It has also to be noted that the SSB in AVL SARvatore and in the official Sentinel-3 data was derived empirically from Jason-2 data. In the AVL SARvatore dataset, SSB data were missing in some rare occurrences further reducing the dataset for comparison. This has, however, a negligible impact on the results. Although SAMOSA+ and ALES + SAR perform better than SAMOSA2 in the close vicinity to the coast, the accurate retrieval of SSH near the coastline remains challenging. The coastal processes and complex coastlines both disturb the altimeter footprint near the coast and alter the sea level variability to an extent where the altimeter retrieved SSH further away might not represent the dynamics near the coast accurately.

**Table 2.** Statistics of the validation of ALES + SAR, SAMOSA+ and SAMOSA2 sea level retrievals against tide gauge data: correlation, root mean square error and number of paired altimetry and tide gauge measurements. Altimetry data are grouped according to their distance to the coast (0–3 km and 3–10 km). Valid ALES + SAR retrievals are selected using a quality flag based on classification, applied to distinguish returns from leads among sea ice, and fitting error. The same statistics derived using SAMOSA2 sea level retrievals on the same points are shown for comparison. SAMOSA+ retrievals are filtered using the same quality flag based on classification and their own misfit quality flag.

| Dataset | Correlation [*r*] | RMSE [*m*] | Number of Points |
|---|---|---|---|
| ALES + SAR (0–3 km) | 0.50 | 0.35 | 4826 |
| SAMOSA+ (0–3 km) | 0.59 | 0.27 | 2177 |
| SAMOSA2 (0–3 km) | 0.44 | 0.43 | 4735 |
| ALES + SAR (3–10 km) | 0.70 | 0.22 | 4502 |
| SAMOSA+ (3–10 km) | 0.58 | 0.26 | 3978 |
| SAMOSA2 (3–10 km) | 0.38 | 0.49 | 4415 |

In order to consider the performance spatially, the statistics are computed at each TG and results are displayed on the Baltic Sea map for the 0–3 km range (Figure 2) and 3–10 km range (Figure 4). The plots show the RMSE, *r* and the number of pairs obtained by ALES + SAR compared to the single TGs, and the differences with respect to the same statistics obtained when using SAMOSA2 and SAMOSA+. An enlarged map for the Danish

Straits is provided in Figures 3 and 5. Note that the statistics at the single TGs were computed only when at least 20 valid altimeter-TG pairs were available and if the result was statistically significant (*p*-value lower than 0.05).

Since SAMOSA2 is evaluated on the same points where valid estimates for ALES + SAR are present, the difference in the number of pairs between ALES + SAR and SAMOSA2 is due to residual outliers found in SAMOSA2 and identified by the comparison with the TGs (see Section 2.4).

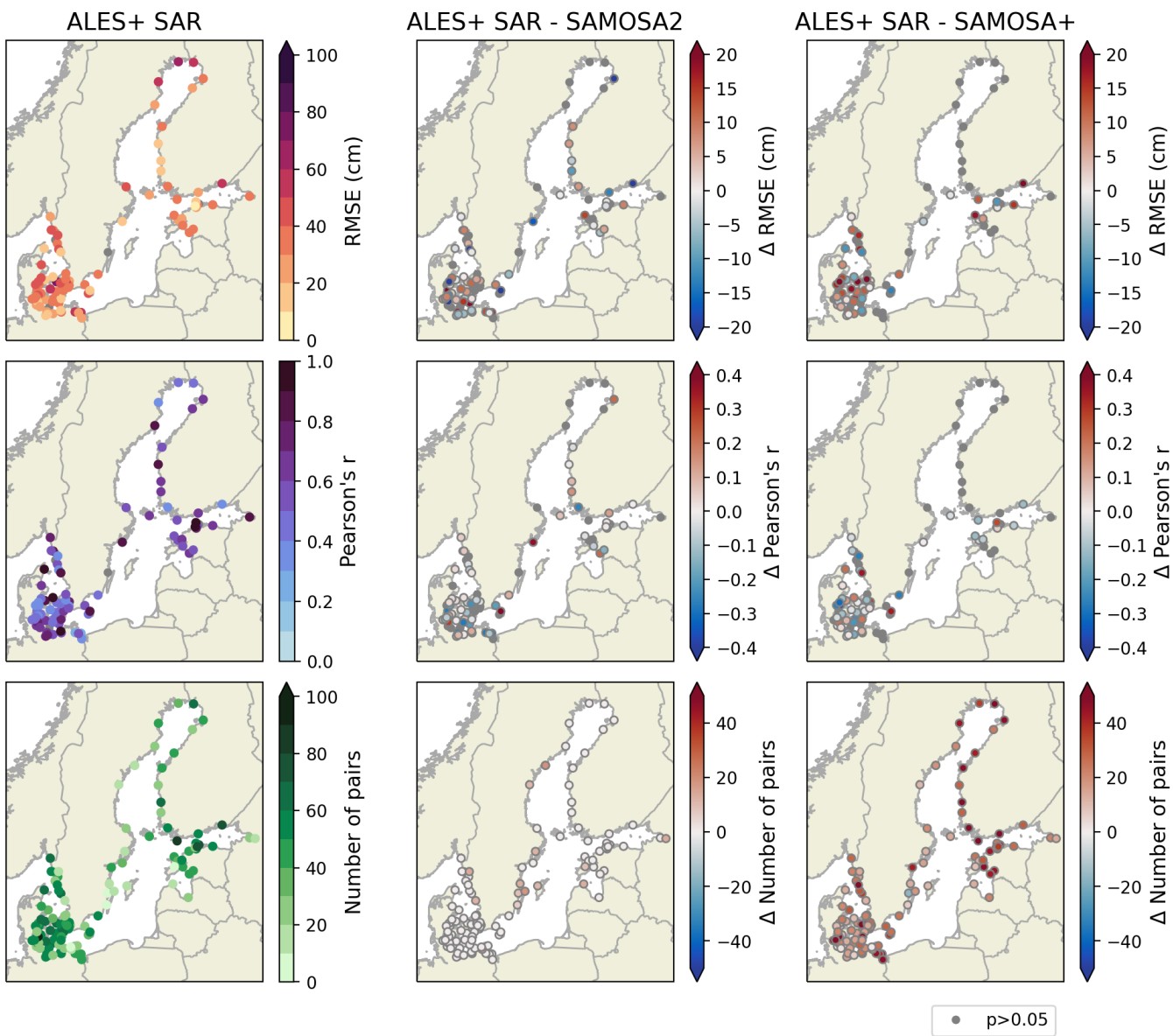

**Figure 2.** Map of root mean squared error (RMSE) and correlation (*r*) computed against the TGs for ALES + SAR, SAMOSA2 and SAMOSA+ (considering altimetry points from 0 to 3 km from the coast). The results are displayed for ALES + SAR (left panel) and then for SAMOSA2 and SAMOSA+ as the difference from the ALES + SAR results (middle and right panels respectively). The number of pairs denotes the amount of comparable altimeter and TG sea level measurements and is reported for SAMOSA2 and SAMOSA+ as a difference from ALES + SAR results in the left panel. Grey dots denote not enough good data to form time series or statistically insignificant correlation (*p*-value > 0.05).

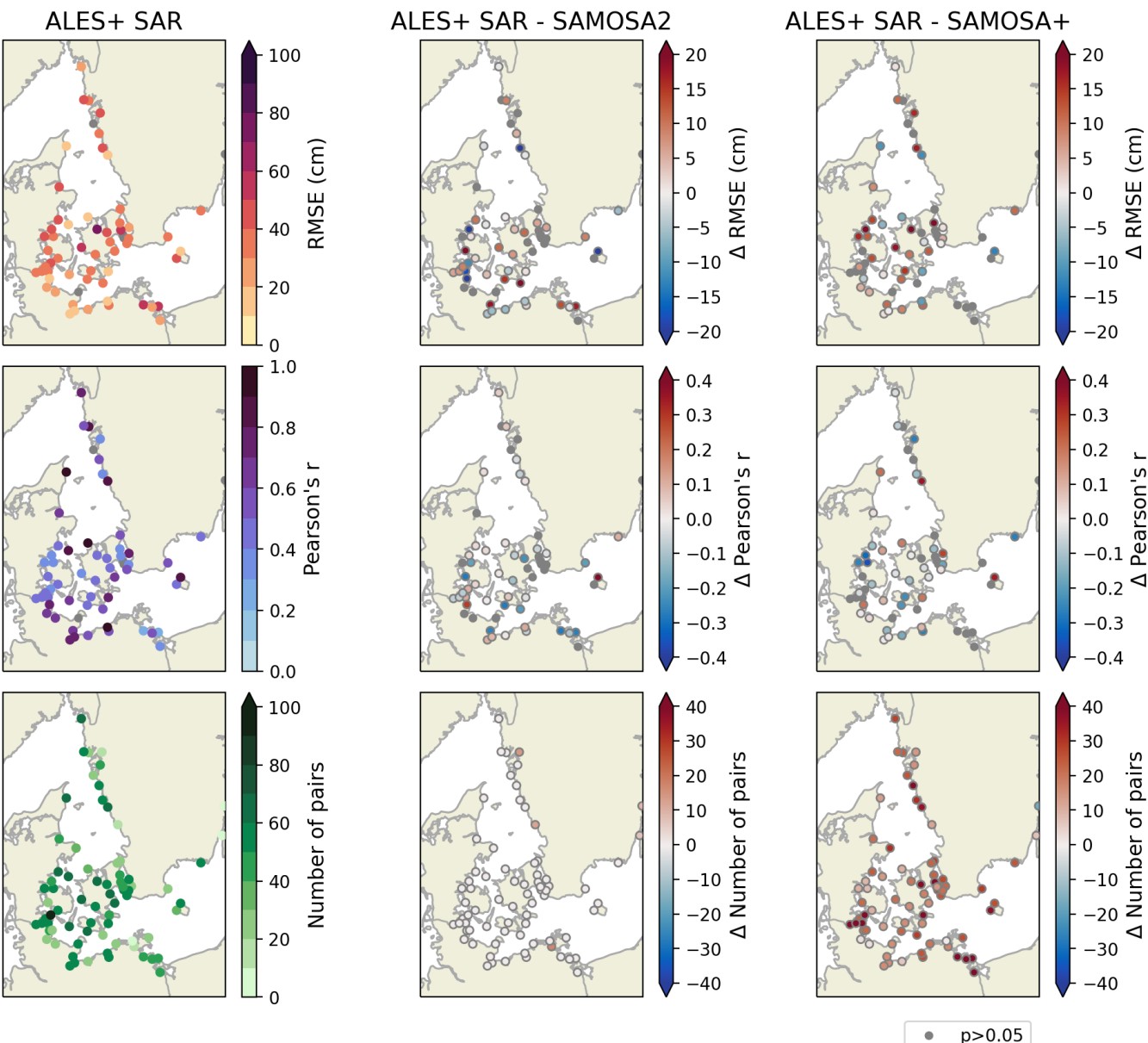

**Figure 3.** Zoom of Figure 2 in the Danish Straits.

We note again that ALES + SAR is able to produce more valid pairings than SAMOSA+, particularly in the 0–3 km band in the Danish Straits. In this region, the sea level could vary locally, on short timescales, for example due to the interplay of coastline and wind direction. The sea level variability at very short scales causes the performances of the altimeters to assume on average lower values than in the rest of the domain. This is nevertheless also the region in which SAMOSA+ shows the best correlation and lowest error compared to the other retracking solutions, although at the cost of a much lower number of valid data. The opposite happens in the 3–10 km band, in which ALES + SAR shows the best performances. ALES + SAR is indeed more able to deal with coastal perturbation in the waveforms when they are further away from the leading edge (see Figure 1b). The closer the perturbation is to the leading edge, the less chances the subwaveform retracker has to correctly fit it.

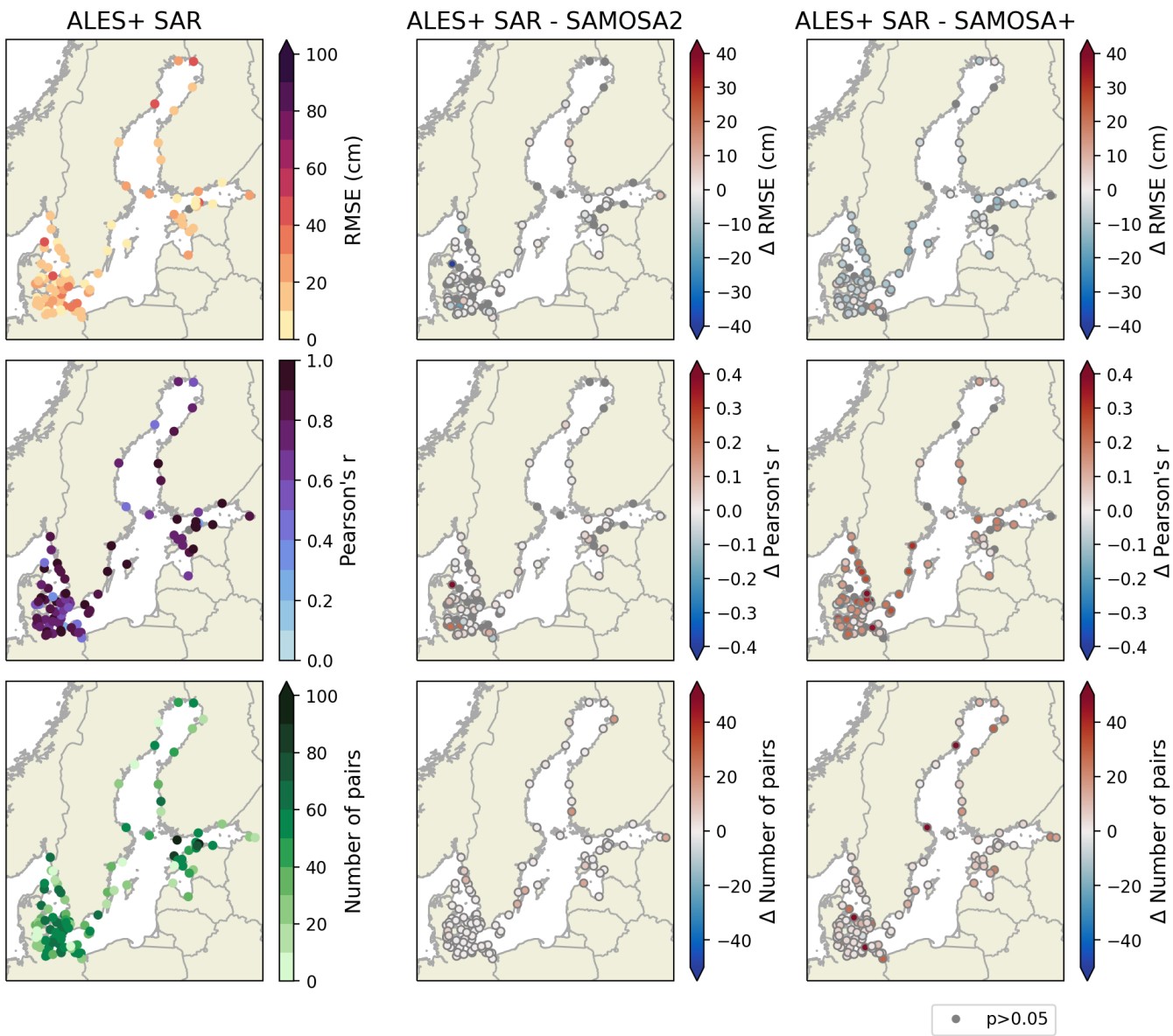

**Figure 4.** Map of root mean squared error (RMSE) and correlation (*r*) computed against the TGs for ALES + SAR, SAMOSA2 and SAMOSA+ (considering altimetry points from 3 to 10 km from the coast). The results are displayed for ALES + SAR (left panel) and then for SAMOSA2 and SAMOSA+ as the difference from the ALES + SAR results (middle and right panels respectively). The number of pairs denotes the amount of comparable altimeter and TG sea level measurements and is reported for SAMOSA2 and SAMOSA+ as a difference from ALES + SAR results in the left panel. Grey dots denote not enough good data to form time series or statistically insignificant correlation (*p*-value > 0.05).

Also, the Bothnian Bay (northernmost and easternmost parts of the basin) show lower performances in the case of the physical retrackers. This is likely to be caused by lower accuracy of the physical retracking of leads among sea ice. The seasonal ice cover in the Bothnian Bay starts forming from the coastline near Kemi in late October [31]. Full sea ice coverage is usually reached by mid-January and sea ice persists until late May [31]. As such, for a major part of the year sea level retrievals in the region depend on leads and other openings in the sea ice. Accordingly, approximately 40% of the data points used for the comparison with the TGs in the northern Bothnian Bay were acquired between the months of November and March. Thus for the Bothnian Bay, the retracking solution becomes increasingly more important in retrieving accurate sea level.

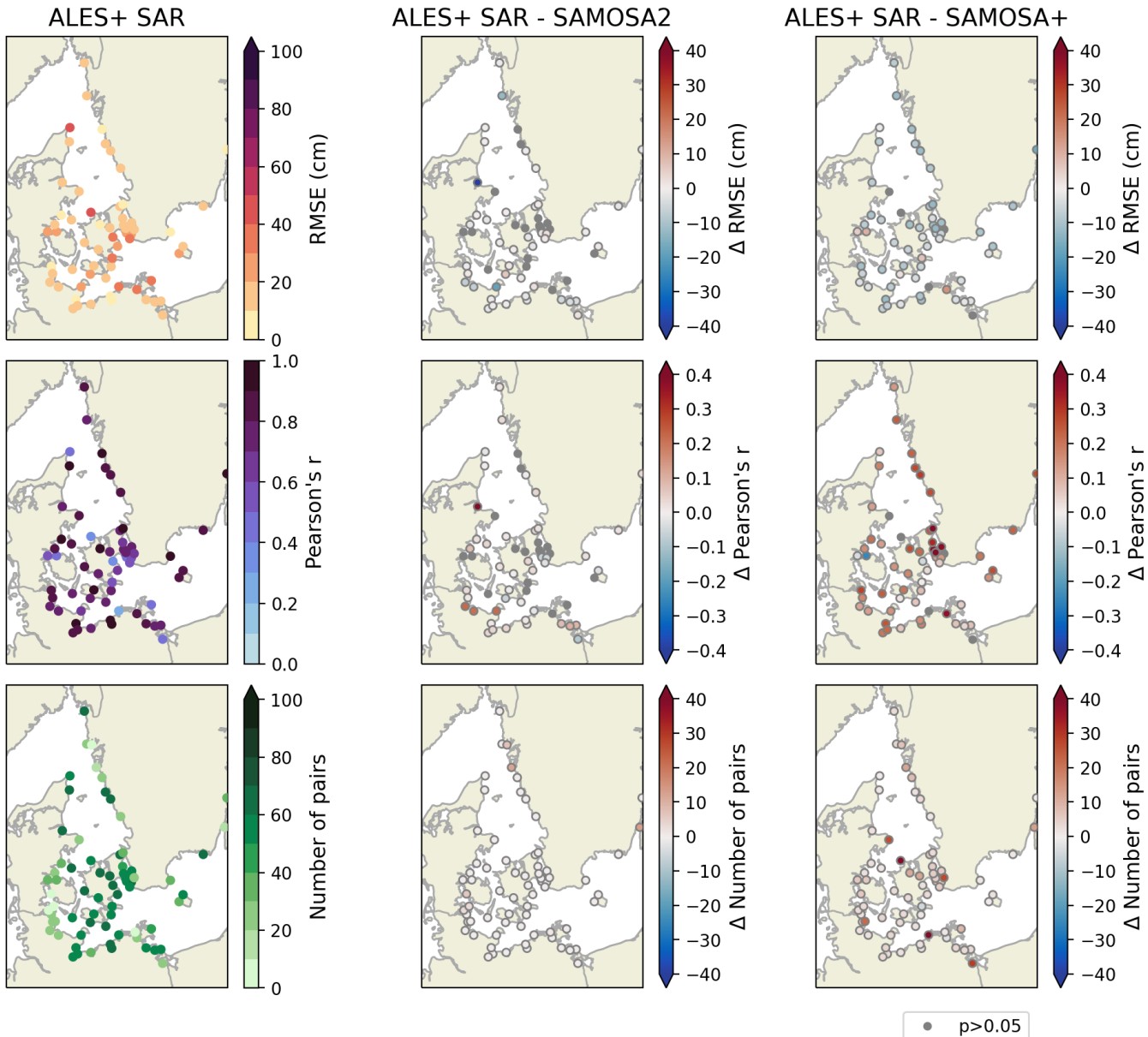

**Figure 5.** Zoom of Figure 4 in the Danish Straits.

*3.3. Crossover Analysis*

When analysing the number of valid single-satellite crossovers (as defined in Section 2.5), it turned out that ALES + SAR has about 1% less valid crossover differences than SAMOSA2 (on average, 188 less values out of 20,554 per cycle). The standard deviations of the crossover differences are generally smaller for ALES + SAR than for SAMOSA2. Over all cycles, the mean standard deviation for SAMOSA2 is 1.66 cm and reduced to 1.55 cm for ALES + SAR. This is an average improvement of 1.1 mm (about 6%) and indicates an improved along-track noise for the ALES + SAR retracker. The mean of all crossover differences as well as its standard deviations (i.e., geographical spread) are larger for ALES + SAR than for SAMOSA2, especially for open ocean areas. Averaged over the entire period and the whole globe, ALES + SAR increases the absolute crossover differences from 5.07 to 5.27 cm and the standard deviation from 7.58 to 7.77 cm (Table 3). However, the differences decreases with smaller distance to the coast. Within coastal regions (up to 10 km from the shore), ALES + SAR outperforms SAMOSA2 by 0.5 mm in mean absolute

crossover differences and 0.8 mm in standard deviation. All numbers from the SXO analysis are summarised in Table 3.

**Table 3.** Results of SXO analysis: mean of crossover differences (mxo), mean of absolute crossover differences (maxo), standard deviation of crossover differences (stdxo), averages standard deviation of crossovers (msxo) [all in cm], as well as number of SXO points. Statistics are given for all crossovers (first two rows) as well as for coastal points only (closer than 10 km from the coast).

| Dataset | mxo | maxo | stdxo | msxo | Number of SXO |
|---|---|---|---|---|---|
| ALES + SAR (all) | −0.18 | 5.27 | 7.77 | 1.55 | 657,731 |
| SAMOSA2 (all) | −0.08 | 5.07 | 7.58 | 1.66 | 663,743 |
| ALES + SAR (<10 km) | −1.62 | 9.60 | 16.79 | 1.30 | 432 |
| SAMOSA2 (<10 km) | −1.51 | 9.65 | 16.87 | 1.47 | 429 |

To detect possible systematic error effects due to the empirical retracker ALES + SAR, a multi-mission crossover analysis is performed. As Figure 6 shows, a significant bias of 9.04 ± 0.25 cm with respect to Jason-3 is observed. Intermission biases are routinely found especially among missions covering different orbits and they are dependent on all processing steps, including retracking [20]. Such biases are corrected in the generation of multi-mission sea level products [32]. More problematic is the case of a clear drift against a reference mission such as Jason-3, since a drift would affect the computation of sea level trends. However, for ALES + SAR no drift can be observed, as the mathematical trend of 0.33 ± 0.42 mm/yr is not statistically significant.

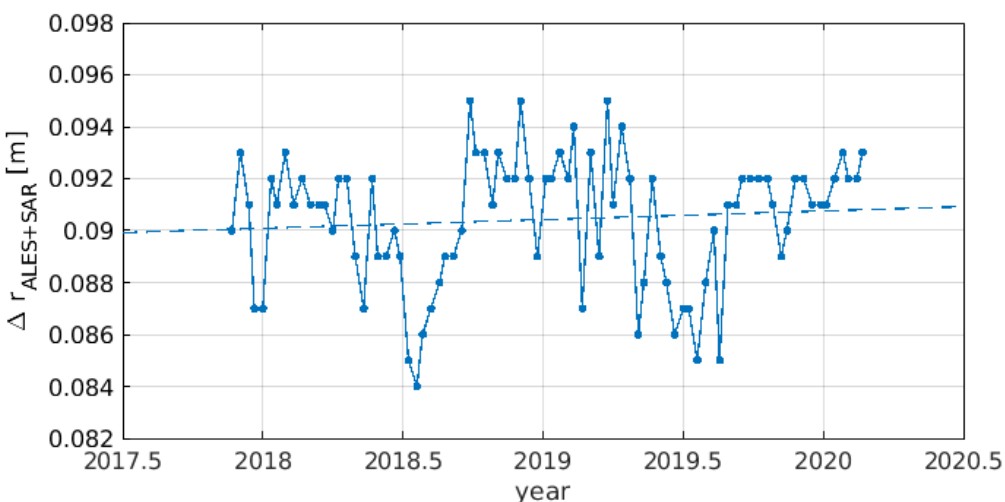

**Figure 6.** Range bias of Sentinel-3A (ALES+SAR retracker) with respect to Jason-3 (per 10-day Jason-3 cycle) as estimated by a multi-mission crossover analysis.

### 3.4. Gridded Sea Level Variance

The gridded SLAVA generated using the ALES+SAR and the SAMOSA2 retrackers is presented and compared in Figure 7. The variances of both retrackers highlight well-known difficulties of retrieving accurate observations from satellite altimetry in regions of high mesoscale variability, which is visible here in the western boundary currents. Hart-Davis et al. [33] and Cancet et al. [34], for example, have shown the difficulty satellite altimetry-derived surface products have in accurately representing western boundary currents compared to in-situ observations. When contrasting the two products, the differences provide insight into the overall and regional performances of the two retrackers. To provide a suitable comparison the scaled variance differences are presented in Figure 7c, with the scaled differences chosen to account for the differences seen in regions of high variances. The difference between these two products for the global ocean, excluding the coastal

region, indicates that the SAMOSA2 retracker had a mean-variance that was 0.99 cm$^2$ smaller than the ALES + SAR retracker. Generally, the two retrackers show similar results particularly in the central to northern Pacific and the Atlantic Ocean. Bigger differences occur in the south Pacific Ocean and, particularly, in the northern Indian Ocean. A possible reason for this is that the SSB of ALES + SAR has been derived only using a limited regional dataset to find a parametric relation with a simple model. This means that specific sea state regimes (for example frequent high sea states typical of the Southern Ocean) are still poorly represented when generating the SSB correction.

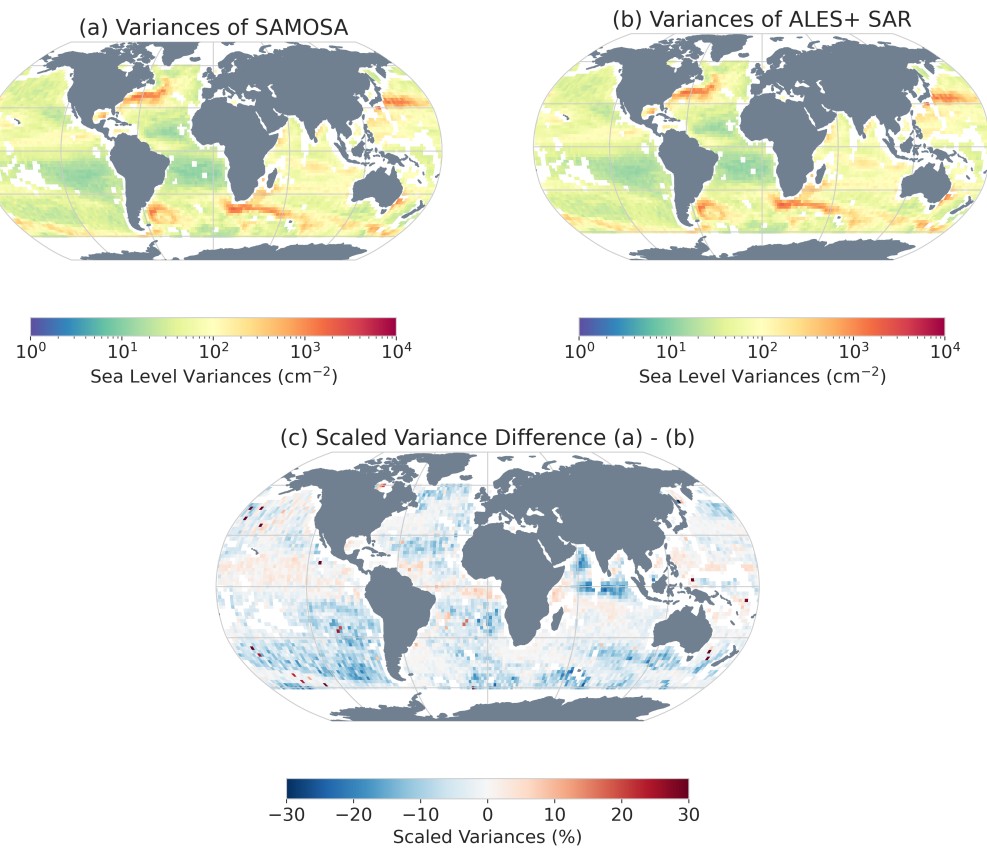

**Figure 7.** The gridded SLAVA from the (**a**) SAMOSA2 and (**b**) ALES+SAR retrackers respectively as well as (**c**) the scaled differences between the respective variances. The coastal grid cells, i.e., grid cells within two degrees of the coast, have been removed due to the two-degree grid size not giving an accurate representation of the coastal variances.

The standard deviation of the sea level estimations for both retrackers relative to the distance to the coast is presented in Figure 8. The standard deviations of both retrackers are the highest closest to the coast and decrease further from the coast. Within the first 10 km, the SAMOSA2 and ALES+SAR retracker show a mean standard deviation of 19.70 cm and 17.38 cm respectively. This difference is explainable since SAMOSA2, as opposed to ALES + SAR, is not a coastal retracker: the global improvement in precision using ALES + SAR in the coastal zone confirms the results observed locally in terms of accuracy (i.e., with respect to in-situ data) in Section 3.2. Although it would be interesting to compare these statistics with SAMOSA+, we do not have global availability of the latter within this study.

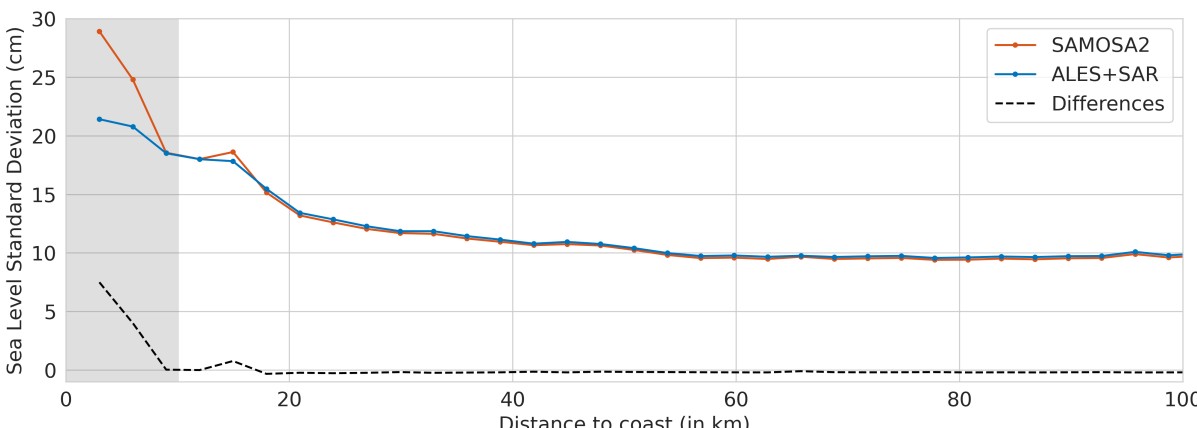

**Figure 8.** The 3-km gridded standard deviation of sea level observations obtained using the SAMOSA2 and the ALES+SAR retrackers. The greyed out area is designated to show the first 10 km of the comparison.

### 3.5. Spectral Analysis

As a final check on the reliability of ALES + SAR, the Power Spectral Density (PSD) spectrum of the sea surface heights is compared with the one obtained using the SAMOSA2 ranges from the original EUMETSAT product in Figure 9. The spectral slope observed is comprised between $k^{-2}$ and $k^{-3}$, which is in line with the global average observed in previous studies [35]. ALES + SAR PSD spectrum, as in the case of SAMOSA2, notably reduces the "spectral hump" characterising the altimetry data from conventional altimetry for scales ranging from 30 to 100 km [36]. Moreover, ALES + SAR reaches a lower noise floor than SAMOSA2. This "denoising" effect has been already demonstrated by [30] and is connected to the reduction of the intra-1Hz correlated errors between the parameters estimated during the retracking, which in this case is attenuated by the application of the SSB correction at 20 Hz.

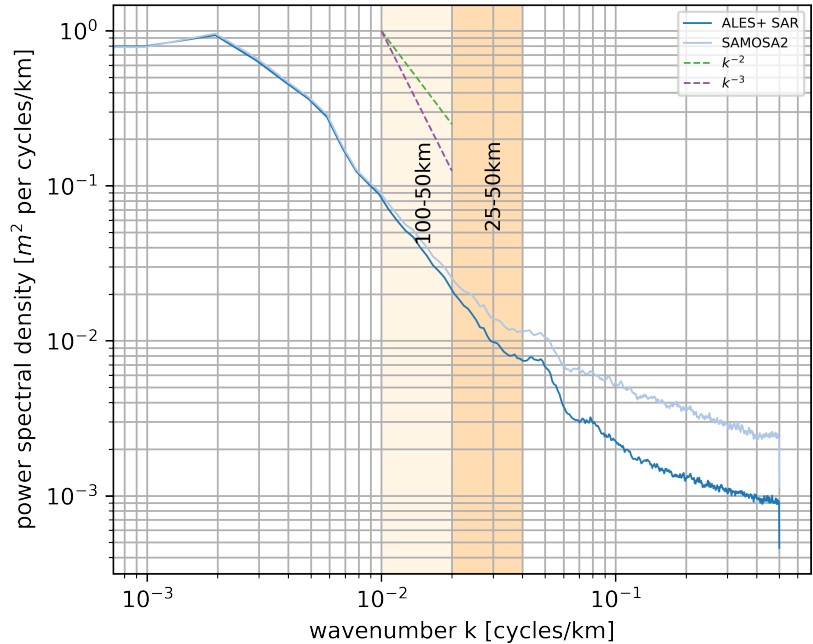

**Figure 9.** PSD spectrum of sea surface heights computed using ALES + SAR and SAMOSA2. The spectrum is based on 8 full cycles (cycles 30 to 37) over the global ocean. To avoid outliers, points located closer than 20 km to the coast and sea level anomalies exceeding 2 m in absolute value are excluded.



## 4. Conclusions

ALES + SAR, an empirical retracker developed in the framework of the ESA Baltic SEAL, has been validated to evaluate its capability in estimating sea level from signals collected with the Sentinel-3A SAR altimeter. In the Baltic Sea, data based on ALES + SAR perform better in the 3–10 km range from the coast than the products based on the current baseline of physical retrackers. This is true not only in the comparison against SAMOSA2, which is designed as an open ocean algorithm, but also in the comparison against SAMOSA+, which is specifically designed to tackle challenging areas such as the coastal zone and those that are covered with sea ice. In the 0–3 km range from the coast, SAMOSA+ performs better than ALES + SAR and SAMOSA2, in terms of correlation and RMSE, but its misfit quality flag reduces the number of points down to a half when compared to the number of points achieved using the ALES + SAR quality flag. The use of the ALES + SAR empirical solution does not negatively affect the drift in range measurements and produces a spectral variability that is at least of the same quality as the current baseline. For the first time considering empirical retrackers, a specific SSB solution was developed. Its impact is shown by the variance decrease, which is observed in the crossover analysis.

In conclusion, we generally agree that physical retrackers are to be preferred since they show comprehensive knowledge of the interaction between the radar signal and the ocean. Nevertheless, this study demonstrates that ALES + SAR is a valid empirical alternative. ALES + SAR notably increases the quality and the quantity of the range retrievals for coastal and sea ice applications, while maintaining the quality of the standard product in the open ocean.

**Author Contributions:** M.P. designed the study and wrote most of the manuscript. M.P. is the author of the ALES + SAR retracking algorithm and of the sea state bias correction used in this study. D.D. was responsible for the crossover analysis and M.G.H.-D. for the variance analysis. F.S. performed the spectral analysis. L.R. and J.S. were responsible for the validation of the TG data. M.R. and J.B. supported the validation activities and provided the SAMOSA+ data F.L.M. is the author of the classification algorithm. C.S. and F.L.M. were responsible for the altimetry database organisation. All authors have read and agreed to the published version of the manuscript.

**Funding:** Part of this research was funded by the ESA Baltic+ Sea Level project (ESA AO/1-9172/17/I-BG - BALTIC +, contract number 4000126590/19/I/BG) and by the ESA Sea Level Climate Change Initiative.

**Data Availability Statement:** SAMOSA+ ranges and SSB corrections are obtained using the SARvatore for Sentinel-3 service of the European Space Agency's Altimetry Virtual Lab (AVL) platform hosted on EarthConsole (https://earthconsole.eu), using the "Sentinel-3" profile preset. Global retracked ALES+ SAR data can be requested from (https://openadb.dgfi.tum.de/). Baltic SEAL data are available at (http://balticseal.eu/outputs/).

**Acknowledgments:** The European Space Agency supported this work through the Baltic + Sea Level project (ESA AO/1-9172/17/I-BG - BALTIC +, contract number 4000126590/19/I/BG) and the Sea Level Climate Change Initiative.

**Conflicts of Interest:** The authors declare no conflict of interest.

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
