# Peer review of "Validation of an Empirical Subwaveform Retracking Strategy for SAR Altimetry"

_remotesensing, doi:10.3390/rs14164122_

Round 1
Reviewer 1 Report
Dear Authors
Your paper is well-written and clear refering to an interesting issue in satellite altimetry close to the coastal regions. I recommend a moderate revision to your paper and I hope that you find my comments useful:
1. In your data analysis, you have cleaned data several times. For example, the points closer than 20 km to the coast and sea level anomalies exceeding 2 m as outliers. Please explain why 20 km in the paper and based on what criterion you considered larger sea level anomalies than 2 m as outliers. You have also done similar cleaning in your work, but you need to justify them. Which statistical test have you applied to detect and remove ourliers.
2. Figures is not clear, ALES+SAR is fine, I understood that the left panels show the RMSE, correlation of differences between TG and ALES+SAR times series, but I think the middle and right panels are solely RMSE and correlation between ALES + SAR and SAMOSA2 and SAMOSA+ not the tide guage time series. This should be clarified more. I do not undertand the numer of pairs in these panels either. I think they need to be clarified more.
3. Again, Figures p-value > 0.05, you have not discussed any statistical test in you paper, no discussion about the confidence interval. p-value of 0.05 means that you have done a test based on a confidence interval of 95%. Explain what you mean by p-value > 0.05. Do you mean you have computed the RMSE and r in the acceptance region of the statistical test? Is the test one-tailed or two-tailed?
4. Figures a histogram with a normal distribution curve on it will show the normallity of the differences. Being in acceptance region is necessary but not sufficient for normality.
5. Figure 6, radial error of what? radial error of Sentinel 3A is not clear. please explain this further ?
6. Figure 7, I think performing a F-test is more suitable than showing the differences of the standard deviations in precentages.
7. Figure 9 shows that the ALES+SAR signal is weaker in high frequencies than SAMOSA2, but you claim that ALES+SAR has lower signal to noise ratio. I do not see that in this figure, I can conclude the opposite. Please clarify in the text.
8. I recommend avoiding paragraphs containing, few sentences, please try to merge them into other relevant paragraphs, or rewrite and rephrase them.
9. Please write an introduction text before opening subsections.
10. In the printed version of your paper, I see ? for references, probably, it is a technical issue regarding the version of the PDF opener. Please check it.
Conclusion of the reviewer
I think the paper is suitable for publication and there are enough new material worth publishing in Remote Sensing. I recommend a moderate revision to this paper and I hope that the authors find my comments constructive.
Reviewer 2 Report
Since the early 90s of last century, global sea level data are routinely collected by radars onboard satellites. And then a lot of algorithms have been proposed to retrieve sea levels from the data. However the sea level retrievals from the latest generation of radar altimeters (the SAR altimeters) are still challenging in the coastal zone and in areas covered by sea ice. In this paper, the performances of an empirical retracking strategy (ALES+ SAR) are evaluated in the Baltic Sea using in-situ data. The results show that ALES+ SAR performances are of comparable quality to the ones obtained using physical-based retrackers, with relevant advantages in coastal and sea ice areas in terms of quality and quantity of the sea level data. The research is useful and the results can be used for other researches. The paper can be accepted for publishing.
Reviewer 3 Report
The authors of the article create some methods aimed at improving the similarity of field measurement data obtained on contact equipment with remote methods obtained on the basis of satellite systems. Research areas are coastal parts of water areas, sometimes covered with ice. The authors of the article in conclusion speak superlatively about the developed methods. But inside the article, we are talking about the best similarity with full-scale contact data, improved by 1%-6%. With these numbers, such admiration. They are very small and can be erroneous. These indicators are not quite correct. It is necessary, in addition to such a better match, to show the errors of the measuring equipment used in contact methods. Paragraph 3.3 refers to accuracy up to 0.1 mm. I am not sure that the equipment used (in contact measurements) gives such accuracy. I ask the authors to give the accuracy of the settings.
Paragraph 3.2 refers to the influence of the coastline and wind direction on sea level variability. This is not exactly a true statement. Not only these parameters affect the surface variability of sea level. Refractive features of sea level changes and nonlinear processes are not taken into account, especially at shallow water, i.e. at sea depths less than half the wavelength. Please take this into account.
Regarding spectral methods. They are aimed only at evaluating the behavior of the spectrum. Figure 9 is better to redraw, introduce more contrasting lines. Moreover, it is necessary to plot the amplitude-frequency characteristics of the measuring equipment in the figure (it is possible separately). Can their amplitude-frequency characteristics influence the measured data?
Round 2
Reviewer 1 Report
Dear Colleagues
Thank you very much for your collaboration. I will recommend accepting your paper in the present form.
Good luck!
Mehdi Eshagh.
Reviewer 3 Report
The article has been significantly improved. You can recommend it for publication.